

Atmospheric
Chemistry
and Physics



# "Warm cover": precursory strong signals for haze pollution hidden in the middle troposphere

Xiangde Xu[1], Wenyue Cai3[1,2,3], Tianliang Zhao[4], Xinfa Qiu[5], Wenhui Zhu[6], Chan Sun[1], Peng Yan[7], Chunzhu Wang[8], and Fei Ge[9]

[1]State Key Laboratory of Severe Weather (LASW), Chinese Academy of Meteorological Sciences, Beijing, China
[2]National Climate Center, China Meteorological Administration, Beijing, China
[3]School of Geographical Science, Nanjing University of Information Science and Technology,
Nanjing, Jiangsu Province, China
[4]Key Laboratory for Aerosol-Cloud-Precipitation of China Meteorological Administration, Nanjing University of Information
Science and Technology, Nanjing, Jiangsu Province, China
[5]School of Applied Meteorology, Nanjing University of Information Science and Technology,
Nanjing, Jiangsu Province, China
[6]Beijing Institute of Applied Meteorology, Beijing, China
[7]Meteorological Observation Center, China Meteorological Administration, Beijing, China
[8]Training Center, China Meteorological Administration, Beijing, China
[9]School of Atmospheric Sciences/Plateau Atmosphere and Environment Key Laboratory of Sichuan Province/Joint
Laboratory of Climate and Environment Change, Chengdu University of Information Technology,
Chengdu, Sichuan Province, China

**Correspondence:** Wenyue Cai (caiwy@cma.gov.cn) and Tianliang Zhao (tlzhao@nuist.edu.cn)

**Abstract.** Eastern China (EC), located in the downstream region of the Tibetan Plateau (TP), is a large area with frequent haze pollution. In addition to air pollutant emissions, meteorological conditions are a key inducement for air pollution episodes. Based on the study of the Great Smog of London in 1952 and haze pollution in EC over recent decades, it is found that the abnormal "warm cover" (air–temperature anomalies) in the middle troposphere, as a precursory strong signal, could be connected to severe air pollution events. The convection and vertical diffusion in the atmospheric boundary layer (ABL) were suppressed by a relatively stable structure of warm cover in the middle troposphere leading to ABL height decreases, which were favorable for the accumulation of air pollutants in the ambient atmosphere. The anomalous structure of the troposphere's warm cover not only exist in heavy haze pollution on the daily scale, but also provide seasonal, interannual and interdecadal strong signals for frequently occurring regional haze pollution. It is revealed that a close relationship existed between interannual variations of the TP's heat source and the warm cover strong signal in the middle troposphere over EC. The warming TP could lead to anomalous warm cover in the middle troposphere from the plateau to the downstream EC region and even the entire East Asian region, thus causing frequent winter haze pollution in EC region.

## 1 Introduction

In China, mainly over the region east of 100° E and south of 40° N (Tie et al., 2009), PM$_{2.5}$ (particulate matter with an aerodynamic diameter equal to or less than 2.5 μm) has become the primary air pollutant in winter (Wang et al., 2017). Therefore, in September 2013, the Chinese government launched China's first air pollution control action plan, "The Airborne Pollution Prevention and Control Action Plan (2013–2017)" (State Council of the People's Republic of China, 2013). By 2017, about 64 % of China's

cities were still suffering from air pollution, especially the Beijing–Tianjin–Hebei region and surrounding areas (Wang et al., 2019; Miao et al., 2019). Then, in July 2018, the Chinese government launched the second three-year action plan for air pollution control, the "Blue Sky Protection Campaign", which demonstrates China's firm determination and new measures for air pollution control (State Council of the People's Republic of China, 2018). After the implementation of air pollution control action plans, air quality in many regions of China has been significantly improved.

Anthropogenic pollutant emissions and unfavorable meteorological conditions are commonly regarded as two key factors for air pollution (Ding and Liu, 2014; Yim et al., 2014; Zhang et al., 2015). Air pollutants mainly come from surface emission sources, and most of the air pollutants are injected from the surface to the atmosphere through the atmospheric boundary layer (ABL) (Quan et al., 2020). The ABL structures are the key meteorological conditions that influence the formation and maintenance of heavy air pollution episodes (Wang et al., 2015, 2016, 2019; Cheng et al., 2016; Tang et al., 2016).

Most previous studies focused on exploring the impact on heavy air pollution in eastern China (EC) from the meteorological conditions in the ABL. However, the thermodynamic and dynamic structures of the free troposphere can affect the meteorological conditions in the ABL (Cai et al., 2020). The convection and diffusion in the ABL are suppressed by a relatively stable structure in the middle troposphere, leading to ABL height decreases, which were favorable for the formation and persistence of heavy air pollution (Quan et al., 2013; Wang et al., 2015; Cai et al., 2020).

This study investigated whether the thermodynamic structure of the troposphere and its intensity changes can be used as a strong warning signal for the changes in $PM_{2.5}$ concentrations in heavy air pollution and whether this strong signal exists on the timescales of seasonal, interannual and interdecadal changes. In order to explore the interaction between the free troposphere and the ABL and the impact on the heavy air pollution in EC, this study extended the meteorological conditions for heavy air pollution from the boundary layer to the middle troposphere. We identify precursory strong signals for frequent winter haze pollution hidden in the free troposphere in EC.

## 2   Data and methods

CE1 The data used in this study included NCEP/NCAR and ERA-Interim reanalysis data of meteorology, as well as data from surface $PM_{2.5}$ concentration measurements, air temperature observations and L-band soundings, as briefly described below.

The monthly NCEP/NCAR reanalysis data of meteorology with horizontal resolution of 2.5° from 1960–2019 were obtained from the U.S. National Center for Environmental

**Table 1.** Air pollution degrees categorized with surface $PM_{2.5}$ concentrations.

| Air pollution degrees | $PM_{2.5}$ concentration ranges |
|---|---|
| Less serious pollution | $75\,\mu g\,m^{-3} < PM_{2.5} \leq 115\,\mu g\,m^{-3}$ |
| Serious pollution | $115\,\mu g\,m^{-3} < PM_{2.5} \leq 150\,\mu g\,m^{-3}$ |
| More serious pollution | $150\,\mu g\,m^{-3} < PM_{2.5} \leq 250\,\mu g\,m^{-3}$ |
| Most serious pollution | $PM_{2.5} > 250\,\mu g\,m^{-3}$ |

Protection (NCEP, https://www.esrl.noaa.gov/, last access: TS1).

The daily and monthly ERA-Interim reanalysis data of meteorology with horizontal resolution of 0.75° were derived from the European Center for Medium-range Weather Forecasts (ECMWF, https://www.ecmwf.int/, last access: TS2), including air temperature, geopotential height, humidity, wind field and vertical velocity.

The hourly $PM_{2.5}$ concentration data from 2013–2019 were collected from the national air quality monitoring network operated by the Ministry of Ecology and Environment of the People's Republic of China (http://www.mee.gov.cn/, last access: TS3). In addition, we categorized air pollution levels with the surface $PM_{2.5}$ concentrations based on the National Ambient Air Quality Standards of China (HJ633-2012) released by the Ministry of Ecology and Environment in 2012 as shown in Table 1.

We also used the monthly air temperature of surface observation data from 1960–2014 from 58 meteorological observation stations in the plateau area with an altitude above 3000 m, which were archived from the China Meteorological Information Center (http://data.cma.cn/, last access: TS4).

Furthermore, the L-band sounding seconds-level data of Beijing from 2010 to 2019 to were used to calculate the height of the ABL (Liu and Liang, 2010). The height of the ABL top is characterized by the L-band sounding observations at 20:00 (local time is used for this paper). The L-band sounding seconds-level data have undergone quality control before analysis (Zhu et al., 2018) and interpolation was implemented in a vertical direction at an interval of 2 hPa. The L-band detection data provided by the China Meteorological Information Center (http://data.cma.cn/, last access: TS5) contains several automatic observation meteorological elements with a time resolution of 1.2 s and vertical resolution of 8 m. More detailed information can be found in Li et al. (2009) and Cai et al. (2014).

## 3   Results

### 3.1   A precursory strong signal of warm cover in the middle troposphere

In February 2014, a rare persistent air pollution weather process occurred in EC with severe air pollution in more than 50

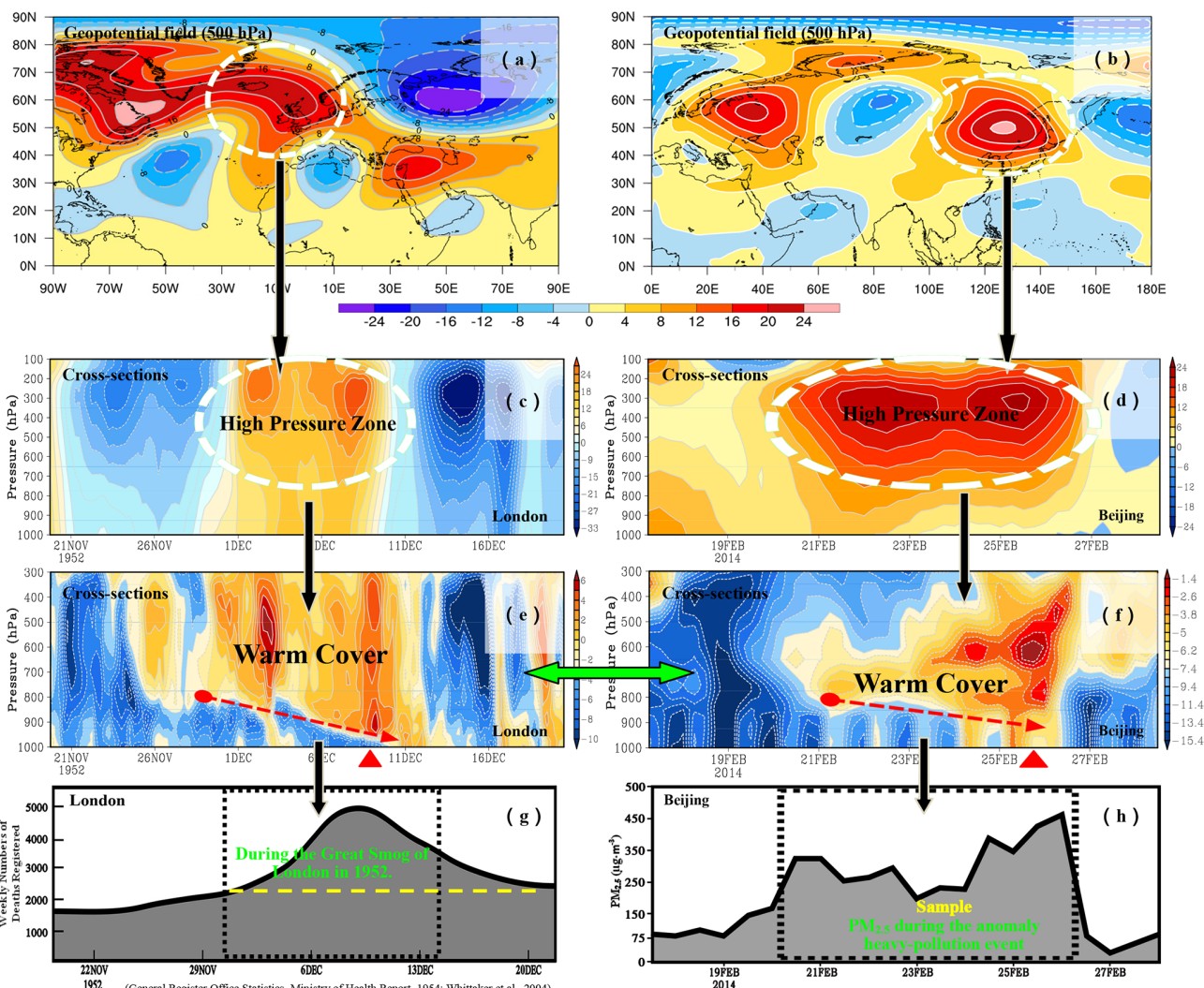

**Figure 1.** Dynamical and thermodynamical structures and air pollution variations. **(a)** Geopotential height anomalies (unit: dagpm) at 500 hPa from 5–9 December 1952 during the Great Smog of London. **(b)** The same as **(a)** but from 21–26 February 2014. **(c)** Time-vertical cross sections of the geopotential height anomalies (unit: dagpm) in the high pressure area (50–70° N; 20° W–10° E) from 20 November to 20 December 1952. **(d)** The same as **(c)** but in the high pressure area (40–63° N; 115–138° E) from 17–28 February 2014. **(e)** Time-vertical cross sections of air temperature anomalies (unit: °C) over London (the red dotted arrow shows the bottom edge of the warm cover during the Great Smog in London) from 20 November to 20 December 1952. **(f)** The same as **(e)** but during the heavy pollution in February 2014 over Beijing. **(g)** Weekly death rate in London prior to, during and after the 1952 pollution episode (General Register Office Statistics TS6, Ministry of Health Report, 1954; Whittaker et al., 2004). **(h)** The variation of surface $PM_{2.5}$ concentrations (units: $\mu g\, m^{-3}$) during the heavy pollution in February 2014 over Beijing.

cities with an impact area of 2.07 million $km^2$. In the Beijing area from 20–26 February 2014, the regional average $PM_{2.5}$ concentration exceed the most serious air pollution level with a peak value of up to $456\,\mu g\, m^{-3}$. In addition, the Great Smog of London in 1952 was attributed to long-lasting and heavy haze pollution under the influence of certain weather systems (Whittaker et al., 2004). To find the precursory strong signals for heavy air pollution events hidden in meteorology, we retrieved the three-dimensional atmospheric dynamic and thermal structures during December 1952 as well as February

2014 by analyzing vertical anomalies of meteorology. There were high-pressure systems moving into London as well as Beijing that stagnated over both areas at 500 hPa geopotential height anomalies, as shown in Fig. 1a and b. During the heavy air pollution events, a high-pressure system over London as well as Beijing gradually strengthened (Fig. 1c and d) and the middle troposphere was characterized by a "warm cover" with upper warming and bottom cooling anomalies in the vertical structure of air temperature (Fig. 1e and f).

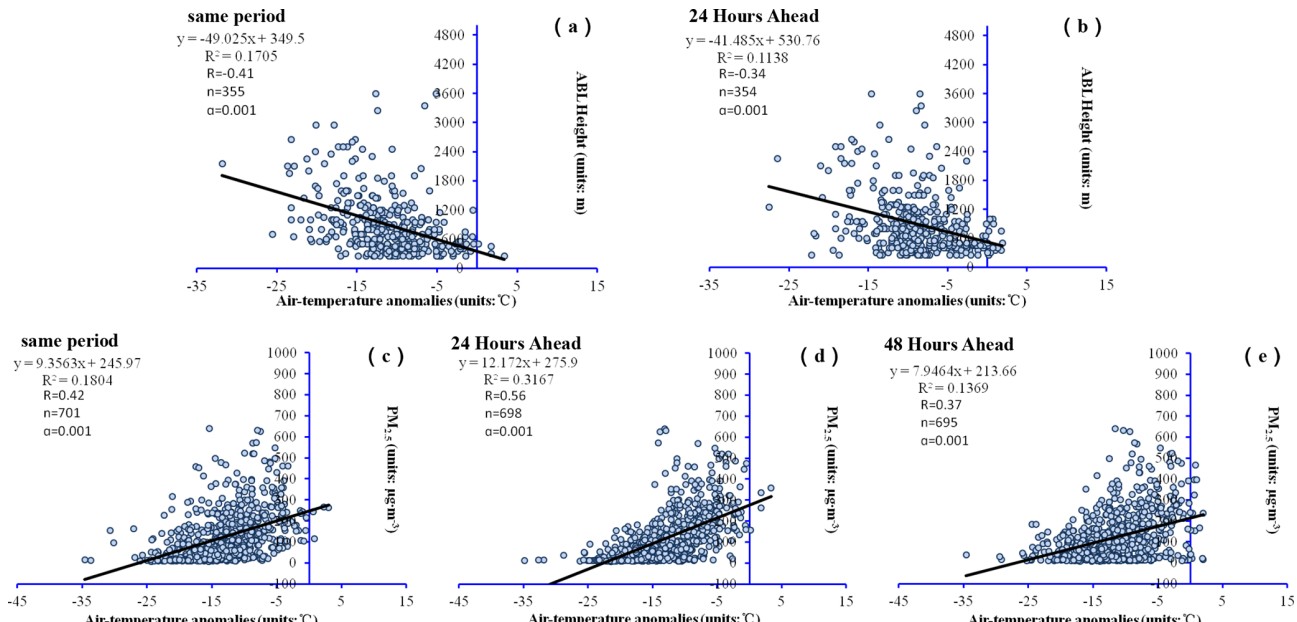

**Figure 2.** The correlations between ABL height and air temperature anomalies in Beijing during winters from 2014–2017 **(a)** in the same period at 800 hPa and **(b)** 24 h ahead at 650 hPa. The correlations between PM$_{2.5}$ concentration and air temperature anomalies in Beijing during winters from 2014–2017 **(c)** in the same period at 850 hPa, **(d)** 24 h ahead at 800 hPa and **(e)** 48 h ahead at 724 hPa.

By comparing Fig. 1a and b, we found that two persistent heavy air pollution events occurred during the maintenance stage of stable high pressure system. During stagnation of the blocking high pressure system, the strength of the center of the geopotential height anomalies in the stable maintenance region of the blocking exhibited a synchronous response to the warm cover above areas (Fig. 1c–f). It can be seen that the local atmospheric thermal structure is significantly modulated by the persistent large-scale anomalous circulation. The subsidence-induced air temperature inversion effect of the blocking high pressure system continuously strengthened the warm cover structure in the middle troposphere, which suppressed the vertical diffusion capacity in the atmosphere (Cai et al., 2020). Moreover, it was obvious that strong signals arising from the thick warm cover persisted during the abnormal air-pollution episode from 5–9 December 1952 in London as well as 21–26 February 2014 in Beijing. It is worth pointing out that the bottom edge of warm cover in the free troposphere declined day by day. During the heavy pollution incident, the warm cover dropped to 900 hPa (Fig. 1g and h). The above analysis shows that in the ABL over London from 5–9 December 1952 and Beijing from 21–26 February 2014, the inversion layer height decreased, which made the ABL structure stable for accumulation of air pollutants. The deep warm cover structures in the middle troposphere acted as a precursory strong signal of the Great Smog of London and Beijing's heavy air pollution.

## 3.2 Effect of warm cover in the free troposphere on the ABL and surface PM$_{2.5}$ variations

During five heavy air pollution episodes over Beijing in December 2015 and 2016 the vertical structures of air temperature anomalies presented the warm cover structure in the free troposphere (see Fig. S1 in the Supplement). During winter 2014–2017, Fig. 2a and b demonstrated the significant negative correlations between the height of the ABL and air temperature anomalies over same period and 24 h ahead in Beijing. The correlation coefficients were 0.41 and 0.34 (99.9 % confidence level), reflecting that the warm cover structure hidden in the middle troposphere with significant strong-signal features is of persistent premonitory significance for the heavy pollution episodes. Figure 2c–e presented the significant positive correlations between PM$_{2.5}$ concentrations and air temperature anomalies over same period and 24 and 48 h ahead in Beijing. The correlation coefficients were 0.42, 0.56 and 0.37 (99.9 % confidence level). Based on the above mentioned results, air temperature anomalies over 24 and 48 h ahead could also reflect that warm cover hidden in the middle troposphere could be regarded as the precursory strong signal for air pollution change. Furthermore, such a stable structure also restricted the vertical transport of moist air from the lower to the middle troposphere for forming secondary aerosols, which could dominate PM$_{2.5}$ concentrations in air pollution over China (Huang et al., 2014; Tan et al., 2015).

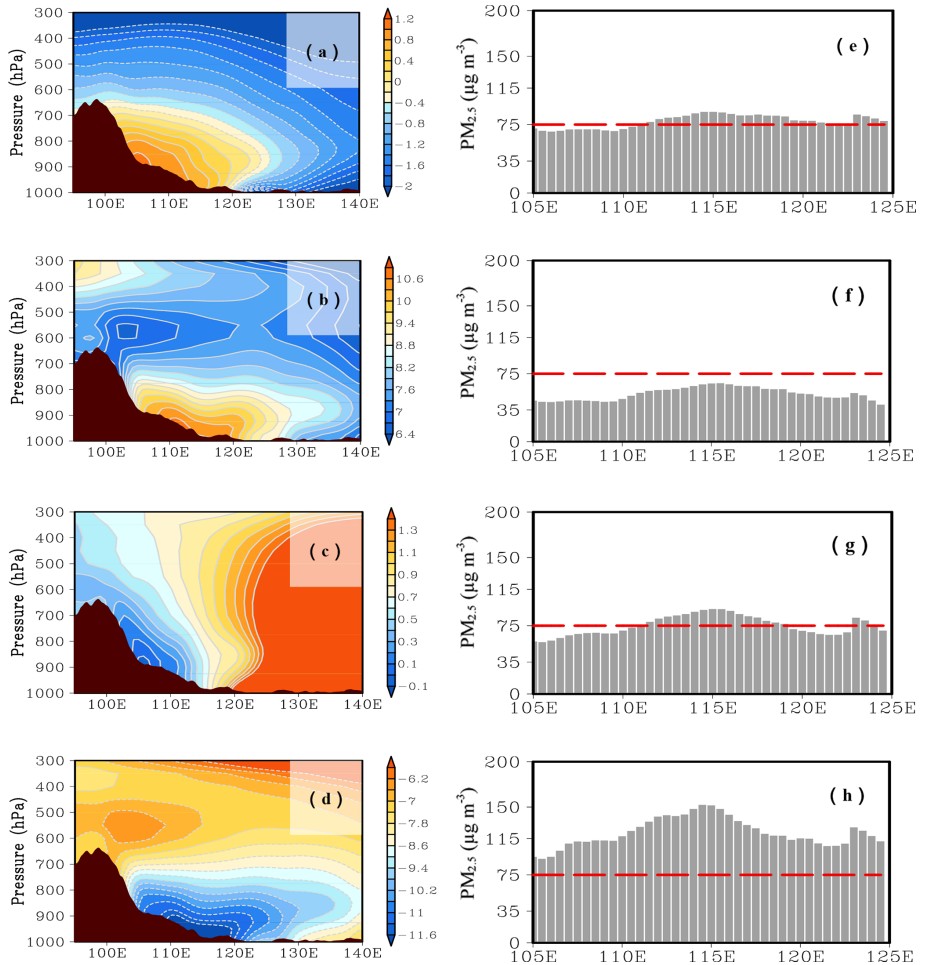

**Figure 3.** Vertical cross sections of **(a–d)** air temperature anomalies (unit: °C) and **(e–h)** the PM$_{2.5}$ concentrations (unit: µg m$^{-3}$) averaged along 25–40° N in spring **(a, e)**, summer **(b, f)**, autumn **(c, g)** and winter **(d, h)** from 2013 to 2018.

### 3.3 Changes of the warm cover structure in the middle troposphere

The warm cover structure of air temperature anomalies in the middle troposphere indicated the intensification of heavy air pollution. The warm cover structure is a precursory strong signal for the frequent occurrence of regional haze events. The air pollution in EC exhibited significant seasonal variations. Our study revealed seasonal differences of the thermal structures in the atmosphere over EC. In spring (Fig. 3a and e) and summer (Fig. 3b and f), the middle troposphere was characterized by an upper cooling and bottom warming vertical structure for less air pollution. When the autumn (Fig. 3c and g) and winter (Fig. 3d and h) arrived, the middle troposphere was characterized by an upper warming and bottom cooling vertical structure, which intensified the air pollution. In autumn, atmospheric thermal structure over EC was marked with a transition between summer and winter (Fig. 3c). The atmosphere condition reversed in winter, a large-scale anomalous air temperature pattern of upper warming and bottom cooling in the middle troposphere appeared from the plateau to downstream EC region and even the entire East Asian region (Fig. 3d). The structure of warm cover in winter was much stronger than that in autumn, and its height of the former was much lower than that of the latter. Therefore, the intensity of air pollution over EC during winter is significantly higher than other seasons (Fig. 3h).

From the perspective of interdecadal variations, our study revealed a close relationship between the frequent occurrence of haze events in EC and the atmospheric thermal structure in the eastern Tibetan Plateau (TP). Furthermore, the thermal structures of the troposphere exhibited distinct interdecadal variations (Fig. 4a–c). A cooling structure was identified in the wintertime air temperature anomalies over the eastern region of the TP from 1961–1980 (Fig. 4a); the upper level of the eastern TP from 1981–2000 showed an upper cooling and bottom warming vertical structure (Fig. 4b). The interdecadal changes of vertical structure reversed from 2001–2018 with a significant warm cover (Fig. 4c). The years 2001–2018 witnessed the highest frequency of haze days

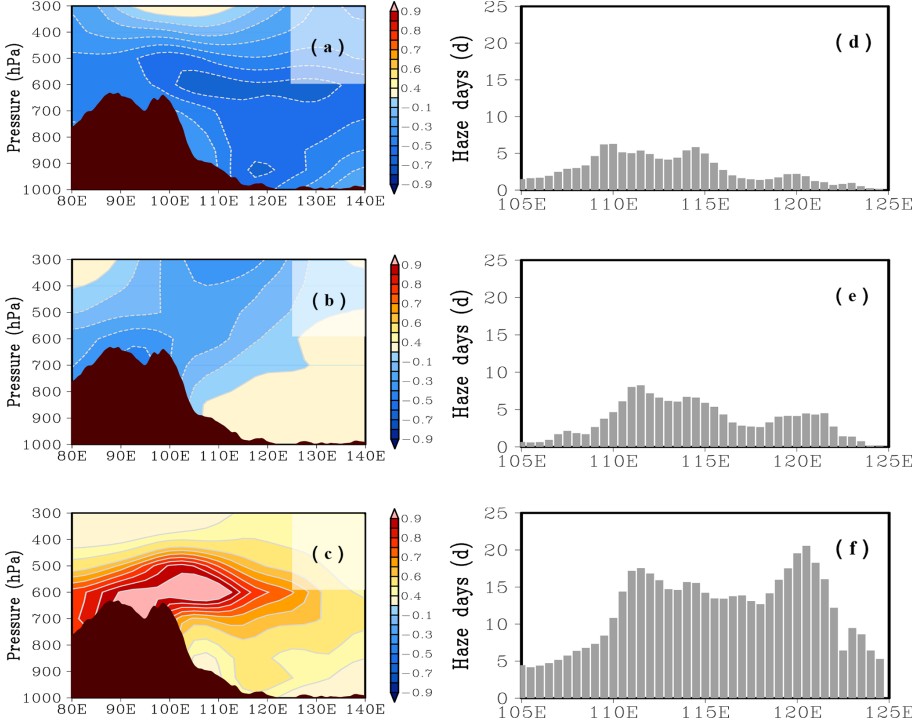

**Figure 4.** Vertical cross sections of **(a–c)** air temperature anomalies (unit: °C) and **(d–f)** the number of haze days averaged along 25–40° N in winter from 1961–1980 **(a, d)**, 1981–2000 **(b, e)** and 2001–2018 **(c, f)**.

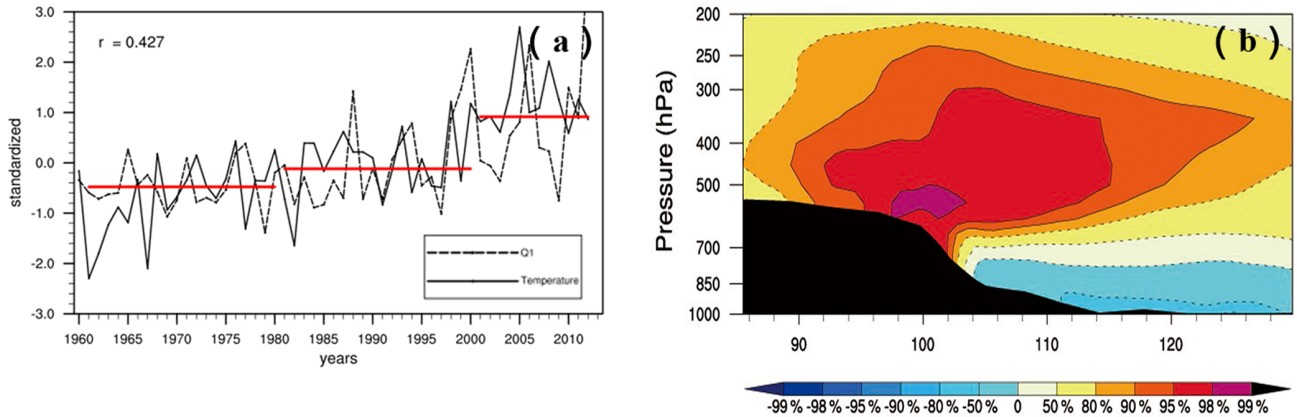

**Figure 5. (a)** Interannual variations of the TP's apparent heat source ($Q_1$) and air temperature of meteorological stations in the TP with altitudes above 3000 m in the winters from 1960–2012. **(b)** Vertical cross sections of the correlations between the TP's apparent heat ($Q_1$) and air temperature latitude averaged along 30–35° N in the winters from 1960–2012.

(Fig. 4f) and 1981–2000 saw a middle-level occurrence of haze days (Fig. 4e), while the lowest frequency of haze days occurred from 1961–1980 (Fig. 4d).

The concept of variations of the tropospheric warm cover has been proposed in this work. Under the background of climate change, it is worth considering whether the variational tendency of the structure of the plateau's heat source induces variations of the tropospheric thermal structure in downstream areas of the Plateau, leading to the interdecadal

variations of the frequency of haze events seen in eastern China since the 21th century. Thermal anomalies of the TP also play an important role in the variations of the frequency of haze events in EC apart from the anthropogenic pollutant emission related to the rapid industrialization of China. The observational and modeling studies have demonstrated that the interannual variations in the thermal forcing of the TP are positively correlated with the incidences of wintertime haze over EC (Xu et al., 2016). The TP induced changes in

atmospheric circulation, increasing atmospheric stability and driving frequent haze events in EC (Xu et al., 2016). In this study, the data analysis concerning the interannual variations of the TP's apparent heat source and air temperature in wintertime at the TP with the altitudes above 3000 m showed that since the 1960s the heat source in areas vulnerable to TP climate change strengthen continuously as the surface temperature increased (Fig. 5a). Furthermore, the TP's apparent heat and air temperature of the middle troposphere over EC presented significant positive correlation (90 % confidence level), which is similar to warm cover structures (Fig. 5b). Therefore, we considered that the warm cover change in the middle troposphere over EC was closely related to the TP's apparent heat and the surface temperature. The TP-induced changes in the thermodynamic structure of the atmosphere provided favorable climatic backgrounds driving air pollution events in EC.

## 4 Discussion and conclusions

Based on the study of the Great Smog of London in 1952 and Beijing's heavy air pollution in 2014, as well as PM$_{2.5}$ pollution over EC, the anomalous warm cover in the middle troposphere was identified as a precursory strong signal for severe air pollution events, which could be attributed to climate change. A stable thermal structure in the middle troposphere, i.e., a warm cover, suppressed the ABL development, which was a key inducement for the accumulation of air pollutants in the ambient atmosphere.

From the perspective of the thermal vertical structure in the troposphere, the abnormal vertical structure in the troposphere during heavy air pollution was investigated. The thermal structure formed by the conventional decline rate of atmospheric air temperature often covers up the anomalous strong signal of the troposphere in the air pollution process, such as the abnormal stable structure with the middle warm and bottom cold in the troposphere with air temperature anomalies. The strong signal of the warm cover of air temperature anomalies in the middle troposphere during heavy air pollution can be described by the method of statistical comprehensive diagnosis analysis.

A large-scale anomalous air temperature pattern of upper warming and bottom cooling in the troposphere appeared from the TP to the downstream EC region and even the entire East Asian region. The frequent haze pollution events in EC since the start of the 21st century happens to be within a significant positive phase in the interdecadal variations of warm cover in the middle troposphere. A close relationship between the TP's heat and the thermal structure in the atmosphere in EC and even the entire East Asian region reflected an important role of the TP's thermal forcing in environment change over China.

*Data availability.* The monthly NCEP/NCAR reanalysis data of meteorology are collected from the U.S. National Center for Environmental Protection (NCEP, https://psl.noaa.gov/data/gridded/data.ncep.reanalysis.pressure.html, last access: TS7). The daily and monthly ERA Interim reanalysis data of meteorology are collected from the European Center for Medium range Weather Forecasts (ECMWF, https://apps.ecmwf.int/datasets/data/interim-full-daily/levtype=pl/, last access: TS8, and https://apps.ecmwf.int/datasets/data/interim-full-moda/levtype=pl/, last access: TS9). The daily and monthly ERA 20C reanalysis data of meteorology are collected from the European Center for Medium range Weather Forecasts (ECMWF, https://apps.ecmwf.int/datasets/data/era20c-daily/levtype=pl/type=an/, last access: TS10, and https://apps.ecmwf.int/datasets/era20c-moda/levtype=pl/type=an/, last access: TS11).

The hourly PM$_{2.5}$ concentration data are collected from the national air quality monitoring network operated by the Ministry of Ecology and Environment the People's Republic of China, which can be found at (https://doi.org/10.5281/zenodo.5372016, Cai, 2021d). The haze days of surface observation data (https://doi.org/10.5281/zenodo.5372006c, Cai, 2021), air temperature of surface observation data (https://doi.org/10.5281/zenodo.5371982a, Cai, 2021) and L-band sounding data (https://doi.org/10.5281/zenodo.5371868b, Cai, 2021) are obtained from the China Meteorological Information Center, China Meteorological Administration All data presented in this paper are available upon request to the corresponding author (Wenyue Cai, caiwy@cma.gov).

*Supplement.* The supplement related to this article is available online at: https://doi.org/10.5194/acp-21-1-2021-supplement.

*Author contributions.* XDX and WYC designed the study. XDX, WYC and TLZ performed the research. WYC performed the statistical analyses. XDX, WYC and TLZ wrote the initial manuscript. TLZ, XFQ, WHZ, CS, PY, CZW and FG contributed to subsequent revisions.

*Competing interests.* The authors declare that they have no conflict of interest.

*Acknowledgements.* The TS12 authors acknowledge the support from the Atmospheric Pollution Control of the Prime Minister Fund, the National Natural Science Foundation of China and the Second Tibet Plateau Scientific Expedition and Research program. We would like to thank Aijun Ding and the two anonymous reviewers, for their suggestions on restructuring the initial draft.

*Financial support.* This research has been supported by the Atmospheric Pollution Control of the Prime Minister Fund (DQGG0104 TS13), the National Natural Science Foundation of China ( TS14 91644223) and the Second Tibet Plateau Scientific Expedition and Research program (STEP, TS15 2019QZKK0105).

*Review statement.* This paper was edited by Aijun Ding and reviewed by three anonymous referees.

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

**Remarks from the language copy-editor**

CE1    **This a case where the editor needs to be asked for the approval. Please give an explanation of why this needs to be changed. We have to ask the handling editor for approval. Thanks.**

**Remarks from the typesetter**

TS1    Please provide this information, day/month/year.
TS2    Please provide this information, day/month/year.
TS3    Please provide this information, day/month/year.
TS4    Please provide this information, day/month/year.
TS5    Please provide this information, day/month/year.
TS6    Does the "Register" belong to the "Ministry"?
TS7    Please provide a reference list entry including creators, title, and date of last access.
TS8    Please provide a reference list entry including creators, title, and date of last access.
TS9    Please provide a reference list entry including creators, title, and date of last access.
TS10   Please provide a reference list entry including creators, title, and date of last access.
TS11   Please provide a reference list entry including creators, title, and date of last access.
TS12   Please note that I left an expression of thanks in this section.
TS13   Is this a grant number?
TS14   Is this a grant number?
TS15   Is this a grant number?