# Peer review of "‘Warm Cover’- Precursory ‘Strong Signals’ hidden in the Middle"

_Atmospheric Chemistry and Physics, 2020_

## Author Response (AR2)

**Reply to Referee 1**

**We are grateful to the referee for the encouraging comments and careful reviews which helped to improve the quality of our paper. In the followings we quoted each review question in the square brackets and presented our response after each paragraph.**

*[Review Comment: This work proposed that abnormal 'warm cover' in the middle troposphere could suppress the convection and diffusion in the boundary layer, leading to haze pollution in Eastern China. It is also indicated that such 'warm cover' is attributed to the warming of the Tibetan Plateau. I think this work well fits the scope of this journal. Overall, this manuscript is well structured but needs more in-depth analysis to further improve this article. Besides, the writing needs to be polished. It is worth being published after addressing the following issues.]*

**Reply:** Thank you for the encouraging comments.

*Major comments:*

*[1. The introduction is too simple and is not sufficient to clearly demonstrate the background and scientific significance of this work. Pre-existing literature on this subject is suggested to be fully reviewed, and a comprehensive introduction ought to be provided in this part.]*

**Reply:** Many thanks for the referee's discussion. For the introduction, we have adjusted it as required and added new content.

**"1 Introduction**

[revised manuscript text omitted]

*[2. The Great Smog of London in 1952 is one of the most well-known air pollution events across the world. Comparatively speaking, the haze in the North China Plain in February 2014 is not that "eye-catching". Why chose this pollution episode for comparison? 2013 Beijing Haze has drawn more attention from both scientific research and public concern.]*

**Reply:** Meteorological conditions in February 2014 were worse than that in January 2013. In February 2014, a rarely persistent air pollution weather process occurred in central and eastern China, this process had caused severe air pollution in more than 50 cities, with an impact area of 2.07 million square kilometers. In the Beijing area

during February 20–26, 2014, the regional average $PM_{2.5}$ concentration exceed the 'most-serious' air pollution level, and with a peak value of up to 456 μg m$^{-3}$.

*[3. It is plausible that 'Warm Cover' may intensify the haze pollution in Eastern China, theoretically. However, as mentioned by the authors, the thermodynamical structure is closely related to circulation, which can significantly influence the regional transport/ventilation of air pollutants. It needs to be clarified whether the anomalous circulation or thermodynamical structure (ABL height decrease) is the main cause of haze pollution. This work only provides correlation and cross-sections of temperature anomalies and $PM_{2.5}$ concentration, both of which are a little too descriptive. More in-depth discussion and some quantitative analysis are suggested to be provided.]*

**Reply:** Your constructive suggestions are greatly appreciated and very helpful for our further study. Due to the limited space of the article, some quantitative analysis will be given in the future, such as the contribution of each meteorological element to polluted weather. This study focused on exploring whether the thermodynamic structure of the troposphere and its intensity changes can be used as a "strong warning signal" for the changes of $PM_{2.5}$ concentration in heavily polluted weathers, and whether this strong signal exists in the time scales of seasonal and interdecadal.

*Minor comments:*

*[1. Line 26: "In addition to"]*

**Reply:** Following this comment, we have adjusted it as required.

*[2. Line 43: Delete "with excessive concentrations of PM2. 5"]*

**Reply:** It has been deleted in the revised manuscript.

*[3. Line 87: the North China Plain. Please check it throughout the manuscript.]*

**Reply:** Following this comment, we have checked it throughout the manuscript.

*[4. Line 88-89: Change to "for the long-lasting and heavy haze pollution". This statement needs to be rephrased. "sulfur-dioxide pollutants" is not appropriate.]*

**Reply:** It has been done in the revised manuscript.

*[5. Line 98: What do you mean by "long heavy air pollution "?]*

**Reply:** Following this comment, we have adjusted it as required.

"persistent heavy air pollution".

*[6. The labels in the contour plot in Fig.3-4 are overlaid and need to be optimized.]*

**Reply:** Following this comment, we have adjusted it as required.

*[7. All the abbreviations should be defined for the first time. Please check throughout the article.]*

**Reply:** Following this comment, we have adjusted it as required.

**Reply to Referee 2**

**We are grateful to the referee for the encouraging comments and careful reviews which helped to improve the quality of our paper. In the followings we quoted each review question in the square brackets and presented our response after each paragraph.**

*[Review Comment: Anthropogenic pollutant emissions and unfavorable meteorological conditions are commonly regarded as two key factors for haze pollution. This study investigated whether the structure of atmospheric thermodynamics in the troposphere and its intensity variation could act as a 'strong forewarning signal' for surface $PM_{2.5}$ concentration variations. It is a very interesting topic and significant for air pollution control. However, I think the current analysis is not sufficient to support the conclusion. Thus, some quantitative estimation and mechanisms illustration is suggested before publication. The detailed reason and suggestions are listed below.]*

**Reply:** Thank you for the encouraging comments.

*[1. Fig 1 and Fig 2 demonstrate the key role of "warm cover" in the haze process. However, the illustration of the connection of "warm cover" with the Tibetan Plateau has lacked. The "warm cover" shown in Figure S1 is below 900 hPa, which is similar to the height of the PBL top. It results in a very stable ABL and further improves the surface $PM_{2.5}$ concentration. However, the "warm cover" induced by Tibetan Plateau is about 600 hPa, which is 4 km. The mechanisms of the impact of "warm cover" in such altitude on PBL is needed to be illustrated in the manuscript.]*

**Reply:** Many thanks for the referee's discussion. We agree with the suggestion. Following this comment, the content of Section 3.3 have adjusted (lines 168-173 and Figure 5) with following

sentences:

"The concept of interdecadal variations of the tropospheric 'warm cover' has been proposed in this work. Under the background of climate change, it is worth considering whether the variational tendency of the structure of the plateau's heat source induces variations of the tropospheric thermal structure in downstream areas of the Plateau, leading to the interdecadal variations of the frequency of haze events seen in Eastern China since the 21th century. Thermal anomalies of the TP also play an important role in the variations of the frequency of haze events in EC apart from the anthropogenic pollutant emission related to the rapid industrialization of China. The observational and modeling studies have demonstrated that the interannual variations in the thermal forcing of TP are positively correlated with the incidences of wintertime haze over EC (Xu et al., 2016). The TP induced changes in atmospheric circulation, increasing atmospheric stability and driving frequent haze events in EC (Xu et al., 2016). In this study, the data analysis concerning the interannual variations of the TP's apparent heat source and air temperature in wintertime at the TP with the altitudes above 3000 meters showed that since the 1960s the heat source in areas vulnerable to TP climate change strengthen continuously as the surface temperature increased (Fig. 5a). Furthermore, the TP's apparent heat and air temperature of the middle troposphere over EC presented the significant positive, which is similar to 'warm cover' structure characteristic (Fig. 5b). Therefore, we considered that the 'warm cover' change in the middle troposphere over EC was closely related to TP's apparent heat and the surface temperature. The TP induced changes in thermodynamic structure of atmospheric provided favorable climatic backgrounds driving air pollution events in EC."

[Figure]

**Figure 5.** (a) TP's apparent heat source (Q1) and air temperature variations with interanual variations of TP's apparent heat source ($Q_1$) and air temperature of meteorological stations in the TP with the altitudes above 3000 meters in the winters during 1960-2014; (b) Vertical cross sections of the correlations between TP's apparent heat ($Q_1$) and air temperature latitude-averaged along 30-35 °N in the winters during 1960-2014.

We have accordingly cited the following article in the revised manuscript:

Xu, X. D., Zhao, T. L., Liu, F., Gong, S. L., Kristovich, D., Lu, C., Guo, Y., Cheng, X. H, Wang, Y. J., and Ding, G.: Climate modulation of the Tibetan Plateau on haze in China, Atmos. Chem. Phys., 16, 1365–1375, https://doi.org/10.5194/acp-16-1365-2016, 2016.

*[2. Fig 3 shows that the "upper warming and bottom cooling" vertical structure in Autumn and Winter favors haze formation. It is interesting. However, the analysis is on the seasonal scale and did not directly support the haze formation on the daily scale.]*

**Reply:** Based on the study of the Great Smog of London in 1952 and the heavy pollution of Beijing in February 2014, it is found that the abnormal 'warm cover' in the middle troposphere connected to both severe air pollution events ( Sect. 3.1 and in Sect. 3.2). This study attempts to explore that whether such the similar structural characteristic of thermodynamic structure, i.e. the abnormal 'warm cover' in the middle troposphere, also exist from the perspective of different time scales, we have further analyzed the $PM_{2.5}$ concentrations and the number of haze days with

seasonal and interdecadal variations of the thermodynamic structures in the atmosphere. We found that the thermal vertical structure of atmospheric showed a 'upper warming and bottom cooling' vertical structure under heavy pollution conditions. The concept of the tropospheric 'warm cover' has been confirmed on the seasonal and climatic scale.

Following this comment, we have added these in the revised Abstract (line 32) as follows:

"The anomalous structure of the troposphere's "warm cover" not only exist in heavy haze pollution on the daily scale, but also provide seasonal and interdecadal 'strong signals' for frequently occurring regional haze pollution."

*[3. Fig 4 compares the interdecadal change of thermal structure in EC and eastern TP with haze days. However, the anthropogenic emissions in EC have increased several times from 1961 to 2018. It is hard to attribute the increase of haze days to the change of TP thermal structure.]*

**Reply:** The pollutant emission with high intensity was the internal cause of frequent air pollution in EC, and the adverse weather conditions were often the key 'inducement' for the accumulation of air pollutants in the atmosphere. Although the variation trends of air pollution in EC depend on the air pollutant emissions,, the air pollution, including its intensity and duration, are closely related to meteorological conditions. Thus, we analyzed the anomalous thermodynamic structure (air temperature anomalies) from the perspective of meteorological conditions, in order to reveal the influence difference of the background field of meteorological conditions on regional atmospheric dispersion conditions. Furthermore, we found that the TP induced changes in thermodynamic structure of atmospheric provided favorable climatic backgrounds driving air pollution events in EC.

*[4. I guess the impact of TP thermal structure on air pollution may cover a large part of EC. Maybe large-scale haze processes could be tried.]*

**Reply:** Yes, the observational and modeling studies have demonstrated that the interannual variations in the thermal forcing of TP are positively correlated with the incidences of wintertime haze over EC (Xu et al., 2016). The TP induced changes in atmospheric circulation, increasing atmospheric stability and driving large-scale haze events in EC (Xu et al., 2016).

[Figure]

Figure 4. Interannual variability in the apparent heat source Q1 (the negative values denote cooling) integrated vertically over the TP and haze event frequency averaged in the CEC in winter (December, January and February) over 1980–2012 and their correlation (upper panel). The haze frequencies (days) averaged in five winters with most positive (lower left panel) and most negative Q1 anomalies (lower right panel) on the TP relative to the mean haze frequency from 1980 to 2012 (Xu et al., 2016).

References:

Xu, X. D., Zhao, T. L., Liu, F., Gong, S. L., Kristovich, D., Lu, C., Guo, Y., Cheng, X. H, Wang, Y. J., and Ding, G.: Climate modulation of the Tibetan Plateau on haze in China, Atmos. Chem. Phys., 16, 1365–1375, https://doi.org/10.5194/acp-16-1365-2016, 2016.

**Supplementary suggestions for revision**

*[Supplementary suggestions for revision: As pointed out by RC2, I still think this manuscript is too descriptive and the majority of the main text is just describing the "warm cover" phenomenon. It does need more quantitative estimation before publication. Or some additional discussion on the implication of this finding can add more scientific significance to this work. In addition, the tense and some English expressions in this manuscript are too confusing and need to be double-checked. English language editing is suggested.]*

**Reply:** Thank you very much to the reviewers. We agree with the comments and suggestions of the reviewers. We have proofread and revised the language of the manuscript, and revised the content of Section 4 of the manuscript:

"Based on the study of the Great Smog of London in 1952 and Beijing's heavy air pollution in 2014, as well as $PM_{2.5}$ pollution over EC, the anomalous 'warm cover' in the middle troposphere was identified as a precursory 'strong signal' for severe air pollution events, which could be attributed to climate change. A stable thermal structure in the middle troposphere, i.e. a 'warm cover', suppressed the ABL development, which was a key 'inducement' for the accumulation of air pollutants in the ambient atmosphere.

From the perspective of the thermal vertical structure in the troposphere, the abnormal vertical structure in the troposphere during heavy air pollution were understood in this study. The

thermal structure formed by the conventional decline rate of atmospheric air temperature often 'covers up' the anomalous 'strong signal' of the troposphere in air pollution process, such as the abnormal stable structure with the middle warm and bottom cold in the troposphere with air temperature anomalies. The 'strong signal' of the 'warm cover' of air temperature anomalies in the middle troposphere during heavy air pollution can be described by the method of statistical comprehensive diagnosis analysis.

A large-scale anomalous air temperature pattern of 'upper warming and bottom cooling' in the troposphere appeared from the TP to the downstream EC region and even the entire East Asian region. The frequent haze pollution events in EC since the start of the 21st century happens to be within a significant positive phase in the interdecadal variations of 'warm cover' in the middle troposphere. A close relationship between the TP's heat and the thermal structure in the atmosphere in EC and even the entire East Asian region reflected an important role of TP's thermal forcing in environment change over China."